# A Robust Deep Model for Classification of Peptic Ulcer and Other Digestive Tract Disorders Using Endoscopic Images

**DOI:** 10.3390/biomedicines10092195

**Published:** 2022-09-05

**Authors:** Saqib Mahmood, Mian Muhammad Sadiq Fareed, Gulnaz Ahmed, Farhan Dawood, Shahid Zikria, Ahmad Mostafa, Syeda Fizzah Jilani, Muhammad Asad, Muhammad Aslam

**Affiliations:** 1Department of Computer Science, Khwaja Fareed University of Engineering and Information Technology, Rahim Yar Khan 64200, Pakistan; 2Department of Software Engineering, University of Central Punjab, Lahore 54000, Pakistan; 3The Institute of Artificial Intelligence and Marine Robots, Dalian Maritime University, Dalian 116026, China; 4Department of Computer Science, Nagoya Institute of Technology, Nagoya 466-0061, Japan; 5Department of Physics, Physical Sciences Building, Aberystwyth University, Aberystwyth SY23 3FL, UK; 6Graduate School of Information Science and Technology, University of Tokyo, Tokyo 113-8656, Japan; 7School of Computing Engineering and Physical Sciences, University of West of Scotland, Glasgow G72 0LH, UK

**Keywords:** deep learning, image classification, supervised learning, imbalanced dataset, WCE dataset, computer-aided diagnosis, BL-SMOTE, class activation

## Abstract

Accurate patient disease classification and detection through deep-learning (DL) models are increasingly contributing to the area of biomedical imaging. The most frequent gastrointestinal (GI) tract ailments are peptic ulcers and stomach cancer. Conventional endoscopy is a painful and hectic procedure for the patient while Wireless Capsule Endoscopy (WCE) is a useful technology for diagnosing GI problems and doing painless gut imaging. However, there is still a challenge to investigate thousands of images captured during the WCE procedure accurately and efficiently because existing deep models are not scored with significant accuracy on WCE image analysis. So, to prevent emergency conditions among patients, we need an efficient and accurate DL model for real-time analysis. In this study, we propose a reliable and efficient approach for classifying GI tract abnormalities using WCE images by applying a deep Convolutional Neural Network (CNN). For this purpose, we propose a custom CNN architecture named GI Disease-Detection Network (GIDD-Net) that is designed from scratch with relatively few parameters to detect GI tract disorders more accurately and efficiently at a low computational cost. Moreover, our model successfully distinguishes GI disorders by visualizing class activation patterns in the stomach bowls as a heat map. The Kvasir-Capsule image dataset has a significant class imbalance problem, we exploited a synthetic oversampling technique BORDERLINE SMOTE (BL-SMOTE) to evenly distribute the image among the classes to prevent the problem of class imbalance. The proposed model is evaluated against various metrics and achieved the following values for evaluation metrics: 98.9%, 99.8%, 98.9%, 98.9%, 98.8%, and 0.0474 for accuracy, AUC, F1-score, precision, recall, and loss, respectively. From the simulation results, it is noted that the proposed model outperforms other state-of-the-art models in all the evaluation metrics.

## 1. Introduction

Gastrointestinal (GI) anomalies like bleeding, pylorus, erosion, ulcers, and polyp are the most frequent kinds of stomach disorders and need extensive medical attention because stomach abnormalities cause several diseases. According to a World Health Organization (WHO) report, stomach cancer was among the top five most frequent malignancies worldwide in 2018 [1]. Traditional endoscopy provides an internal view of the GI tract. Moreover, this traditional method cannot observe the small intestine as it is very long and complex. Moreover, this endoscopy method is inconvenient and painful for patients. Wireless Capsule Endoscopy (WCE) is another innovative GI diagnostic device with a small capsule including a camera and transmitter. In WCE, there is a transmitter inside the capsule device that captures infections of the GI tract and it takes almost 60 thousand images of the GI tract and provides better comfort and causes less pain to the patient.

The GI tract, liver, pancreas, and gallbladder are all parts of the body’s digestive system, which aids in food digestion. It takes digestion to turn food into the nutrients that your body needs for energy, development, and cell repair. However, GI tract diseases are a serious concern to human health. For instance, the second most common cause of cancer-related mortality worldwide is gastric cancer, which is the fourth most prevalent kind of cancer worldwide [2]. WCE is very useful in assessing localized lesions in the large intestine, such as those linked to GI bleeding and ulcers [3]. During the examination phase, a patient ingests a capsule that is driven down the GI system by peristalsis or magnetic fields [4,5]. The WCE travels while taking coloured photos of the GI tract at a frame rate of 2–4 pictures per second [5] and transmits those images to a data-recording device; then, these captured images are inspected by doctors to make a diagnosis.

Ulcers, bleeding, pylorus, erosion, and polyps in the digestive system are the main illnesses identified by the WCE. The images obtained from the WCE are used to detect the anatomical landmarks, pathological abnormalities, and poly removal, which are crucial GI disorders. Offering a variety of images provides a more practical way to diagnose tumours and gastrointestinal haemorrhages, particularly in the small intestine, which is now more accurately examined [6]. Analyzing every picture that was captured from each patient required a lot of time [7]. Moreover, sometimes there is a high similarity rate in different contextual images; that is why the most skilled doctors face challenges that need a lot of time to assess all of the data. The doctor must watch the full film in order, even though the bulk of the images is filled with meaningless information. This frequently leads to misdiagnosis because of incompetence or carelessness [8] among the doctors.

Many researchers from all around the world have created a variety of ML models [9] that have shown notable proficiency in carrying out such automated activities, and they can develop and improve biomedical image analysis. However, Convolutional Neural Network (CNN)-based deep-learning techniques have made significant strides in classification problems recently [10]. This collection of DL models introduces a hybrid CNN and CNN model with histogram stretching [11]. We propose a deep-learning method to diagnose ulcers and other GI tract abnormalities using a big-volume WCE dataset to provide adequate diversity. This is because it is challenging to mathematically describe the great variation in the shapes and features of affected regions in WCE images and deep learning is a powerful tool for accurate feature extraction.

CNN is a system of Input, output, and several hidden convolutional layers that serve as intermediary layers in CNN design. Moreover, it also includes dense layers for training parameters based on convolutional layers, and flattened layers to convert multi-dimension inputs into one-dimensional output [9,10]. These layers carry out operations on the data to discover characteristics unique to the data. The neurons in each layer of the neural network calculate the weighted average of their input, and this average is then passed through a non-linear function known as an activation function [10,11]. DL provides a variety of activation functions such as Rectified Linear Unit (ReLU), and SoftMax is a fundamentally essential part of CNN. For automatic image feature extraction, CNN offers a variety of filters that can help in pattern identification.

As the result of the above discussion, we conclude that WCE offers a non-invasive painless alternative to standard endoscopy for imaging the whole GI tract, including the small intestine. It can be laborious and time-consuming for the doctors to manually diagnose each image in a scenario where the percentage of WCE images with anomalies is just 5–7% of the total WCE images gathered [7,8]. Consequently, a method for automated computer-aided anomaly detection to help clinicians diagnose gastric ulcers is needed [12]. Numerous studies have been undertaken in the past on the topic of GI tract ulcer identification for endoscopic pictures. However, there has not been much research conducted on utilizing WCE pictures to detect other common gastric problems [4,5]. Additionally, these methods use a small dataset with a few classes and segmentation methods to find ulcers. It is challenging to increase detection accuracy using traditional image processing and segmentation techniques.

In this study, we propose a technique for identifying ulcers as well as the most frequent GI tract disorders in WCE images by employing a CNN model that exploits several conventional blocks made up of various deep layers. The suggested approach aims to obtain a precise classification result for more accurate identification of GI tract ailments. These are the primary contributions of the proposed deep-learning model:We propose a robust CNN architecture for GI tract disorders which offers a variety of filters that can help accurately identify the lesion and we also decrease the model complexity by reducing the number of training parameters to achieve remarkable classification results.Due to the class imbalance problem of medical datasets, the accuracy of the CNN models is compromised. We solve this problem by employing an up-sampling algorithm, BL-SMOTE, to generate concoction image samples concerning each class to achieve better accuracy.The Grad-CAM heat-map algorithm is used to depict the visual characteristics of GI tract ailment approaches to reaching classification decisions.Using a variety of evaluation metrics, including accuracy, AUC, precision, recall, F1-score, loss, and quantity of trainable parameters, we also compared our proposed architecture to existing models. It has been found that our method performs better than other cutting-edge models.

This article introduces and familiarizes other researchers with a framework for visualizing the classification process of a deep-learning model. The technique creates several intermediate pictures as a sample image progresses through the convolution layers. The remaining portion of the study is structured as follows: It exposes the opaque nature of CNN models, enabling researchers to comprehend what features a model is extracting and which region of interest (ROI) it is basing a certain class label for a sample on. A summary of the pertinent research is given in Section 2. The methodologies and suggested models for classifying the most common stomach diseases and peptic ulcers are presented in depth in Section 3. The data collection and model components are described in Section 4. Section 5 presents the visualization procedure. The model evaluation with different quality metrics is completed in Section 6, and finally the conclusion is drawn in Section 7.

## 2. Materials and Methods

Only a small number of modalities have been deployed over the years in the field of WCE image analysis with an emphasis on model designs and performance. Due to the challenging process of gathering medical datasets, precise classification of medical imaging is a taxing endeavour [13]. Medical datasets, in contrast to other datasets, are created by qualified professional clinicians and contain confidential and sensitive patient information that cannot be made available to the general public. To save researchers time and provide free access to these datasets to the research community, certain organizations created medical datasets. This research focuses on the automatic identification of GI tract abnormalities and finding, polyp, pylorus, blood fresh, and symptoms of some of the most frequent gastric ailments [14]. A sizable WCE dataset was gathered through real-time patient analysis at a Norwegian hospital. There are 117 videos in the Kvasir-Capsule, from which 4,741,504 picture frames may be extracted. We surrounded data from 14 distinct classes with 47,238 labelled frames that have been medically confirmed by specialists. However, there is another problem that arises while collecting medical datasets; it is impossible to collect data collection with an equal number of samples of healthy and ill patients, medical datasets are intrinsically very unbalanced, and the techniques to tackle this problem are quite challenging themselves [15].

To achieve remarkable classification results utilizing WCE images, researchers use both conventional machine-learning techniques and deep-learning algorithms [15]. It is quite difficult to increase the categorization of illness regions with high levels of accuracy in automated detection and WCE needs cutting-edge deep-learning approaches to improve its analytical age [16]. It is suggested that the AlexNet model can be used to categorize the upper gastrointestinal organs from the images taken in various environments. In terms of classifying upper gastrointestinal anatomical structures, the model has an accuracy of 96.5% [16]. Based on an examination of factorization, the authors suggested a method to speed up the endoscopic screening review process [16]. The single value decomposition sliding window technique is employed. The approach provides a 92% overall accuracy [17].

A four-layer Convolutional Neural Network was suggested by Charisis [18]. as a way to categorize various types of ulcers from extracted WCE frames. By adjusting the model’s hyper-parameters, the test results are enhanced and accuracy of 96.8% is attained. A novel virtual reality capsule was created by Kagan Incetan et al. [19] to mimic and distinguish between normal and pathological regions. For gastrointestinal illnesses and GI tract disorders, this capsule can generate novel 3D pictures. In [20], the author proposed a Gabor capsule network for classifying complex images of the Kvasir dataset. The model achieves an overall accuracy of 91.50%. The wavelet transform with a CNN is proposed to classify gastrointestinal tract diseases and achieves an overall average performance of 93.65% in classifying the eight classes [21].

For the classification of anomalous images, transfer learning-based deep-learning solutions incorporate pre-existing CNN models, such as DenseNet, ResNet, and VGG16, while other solutions use customized CNN models [22,23,24,25]. A few models were put up for the classification of multi-class anomaly images as well as those for the classification of particular anomaly classes, such as a polyp, ulcer, and haemorrhage [26,27]. Ghosh et al. [27] suggested utilizing SegNet and pre-trained AlexNet in a deep transfer learning strategy to categorize the bleeding frames and locate the bleeding locations. Yu et al. [22] suggested a hybrid CNN with diverse classifier functions, such as SoftMax and extreme machine classifiers to increase the classification accuracy of the CNN model.

In DL, there is always scope for improvement and most of the researchers have not been able to achieve remarkable classification performance. Their methodologies and approaches suffer from various hindering factors because they overlook some inherent hurdles of DL models and medical image datasets [15,16,20]. A comparative analysis of state-of-the-art WCE classification models is depicted in Table 1. We are working on incorporating traditional deep and transfer learning models with novel approaches to achieve outstanding results of accuracy greater than 95%. The dataset used in this research is collected from Scientific Data which contains 47,238 labelled image samples of anonymous patients with only WCE images and their respective class label information. It is a multi-class dataset consisting of fourteen different classes including a normal (normal clean mucosa) class and thirteen other classes representing thirteen different abnormal symptoms of the GI tract, namely ampulla of vater, angiectasia, blood fresh, blood hematin, erosion, erythema, foreign body, ileocecal valve, lymphangiectasia, polyp, pylorus, reduced mucosal view, and ulcer. It is a year-old dataset and few researchers have offered their contributions in this duration while obtaining good results by employing several techniques and their combinations.

## 3. The Proposed GI-Detection Model

The world of healthcare has seen a significant revolution because of image processing and artificial intelligence. These days, image processing is used in nearly every field of healthcare for pre-analysis [1,10,13]. Without requiring surgery, doctors may inspect the human body’s organs from the inside during the diagnosing phase. Many medical devices provide a painless and reliable examination for major diseases. Endoscopies need holes in the body, such as the mouth or anus, and are considered minimally invasive procedures [19]. However, two different modalities are used to examine the abnormal condition of the GI tract. The first method is traditional endoscopy in which a long, thin tube is inserted into the GI tract to closely examine a damaged internal organ or affected tissue and this procedure is painful for patients [12,14,17]. The second method is WCE in which a tiny capsule, about the size of a big vitamin tablet, is swallowed during a capsule endoscopy operation. A tiny wireless camera that takes photographs as the capsule travels through the small intestine is embedded inside the capsule and Images are sent to a recorder that is attached to a system [1,2]. Medical experts are not capable of analyzing the number of images generated by capsule devices accurately; it requires a lot of time. A computer can make appropriate inferences from them because a machine that has been trained on a collection of medical picture data may deliver precise conclusions in a matter of seconds [16,18,21]. Computer vision and image-processing algorithms play a crucial role in today’s health care systems. GI tract problems are becoming very common issues because of low-quality foods. The research community plays a notable role in building intelligent automated systems for fast and accurate examination and contributes to day-by-day enhancements [4,5,7].

In this proposed model, the input dataset is pre-processed by utilizing normalization and implementing the crucial step of transforming the categorical data variables before they are supplied to the proposed CNN model. Then, to balance the dataset and address the problem of an unbalanced dataset, we used the Border-Line Synthetic Minority Oversampling Technique (BL-SMOTE) to balance the number of samples in each class [31]. The dataset is then divided into three sections: train, test, and validation. Additionally, there is efficient model training of the suggested one presented in Figure 1. The size of training parameters is smaller in comparison with [18,20,21]. The Grad-CAM heat-map algorithm is used to depict the visual characteristics of GI tract ailments which highlight the features which lead to classification of an image sample.

### 3.1. WCE Dataset Description

There are many datasets of endoscopy images that are present and publicly available on the internet. Most of the datasets are a binary class that expresses a lesser number of infections that are present in the GI tract. The dataset used in this research was published in May 2021 and extracted from scientific data named Kvasir-Capsule [14]. This multiclass dataset has fourteen distinct classes prepared by extracting video frames from WCE videos. Initially, videos were gathered from sequential clinical exams conducted at the Department of Medicine, barium hospital, and Vestre Viken hospital Trust in Norway, which serves 490,000 individuals and covers around 200,000 of them. Utilizing the Olympus endocapsule 10 System [10], which includes the Olympus endocapsule and the Olympus RE-10 endocapsule recorder, the exams were carried out between February 2016 and January 2018. The videos were initially recorded at a rate of 2 frames per second, with a resolution of 336 × 336. The videos were exported using the Olympus system’s export tool in AVI format; it was packed but the export tool modified the frame rate specification to 30 fps.

According to the given details of the dataset, each sample in the dataset available on scientific data is personally verified by clinical professionals. As is known, the dataset contains fourteen distinct classes named class 0 (ampulla of vater), class 1 (angiectasia), class 2 (blood fresh), class 3 (blood hematin), class 4 (erosion), class 5 (erythema), class 6 (foreign body), class 7 (ileocecal valve), class 8 (lymphangiectasia), class 9 (Normal mucosa), class 10 (polyp), class 11 (pylorus), class 12 (Reduced mucosal), class 13 (ulcer). The samples are individual three-channel (RGB) images of 128 × 128 pixels dimension belonging to four different classes [10]. and the number of samples in the normal class is 34,338. The thirteen classes—ampulla of vater, telangiectasia, blood fresh, blood hematin, erosion, erythema, foreign body, ileocecal valve, lymphangiectasia, polyp, pylorus, reduced mucosal, ulcer—have 10, 866, 446, 12, 506, 159, 776, 4189, 592, 55, 1529, 2906, 854 images, respectively, as depicted in Figure 2. The only downside of this dataset is that it is imbalanced, as discussed in Table 1. As can be seen, there are too many image samples in the normal mucosa class concerning other class samples; this will cause an increase in the size of the dataset which may lead to a model overfitting problem. That is why we reduce the number of samples in a normal class to 5927 for a suitable size of the dataset and achieve remarkable classification results. Based on these factors, this dataset is used in our research; it has 18,827 samples in total.

### 3.2. Balanced Each Class Sample Using BL-SMOTE

To overcome the class imbalance problem of the dataset we use the up-sampling technique. To enhance the sampling rate, up-sampling involves adding zero-valued samples between the original samples. In this approach, we use an up-sampling algorithm BL-SMOTE [31] to generate fusion samples concerning each class, as shown in Figure 3. In this algorithm, minority class observations are first classified. If all of the neighbours are members of the majority class, it labels any minority observation as a noise point and ignores it while producing synthetic data. Additionally, it resamples entirely from a small number of border places that are neighbourhoods for both the majority and minority classes. The distribution of samples after employing up-sampling is presented in Table 2.

## 4. Components of Proposed Classification Model for WCE

The details of the proposed deep-learning model for GI disorders and its architecture are given in the next section.

### 4.1. Detailed Architecture of the Proposed System

The CNN architecture is modeled according to the biological structure of the human brain and is especially useful for computer vision tasks including object identification, face detection, image segmentation, and classification.

Its translation-invariant feature meant it was chosen by previously created deep models [26,32,33]. According to the translation or space invariance, a CNN can detect the same feature regardless of where it appears in different pictures. We created a robust custom CNN model in this research to accurately classify critical infections of the GI tract. Our model consists of four convolutional blocks, which comprise a Soft-Max classification layer, two dropout layers, two dense layers, and a Rectified Linear Unit (ReLU) activation function as shown in Figure 4. A detailed summary of the proposed deep CNN model for the classification of stomach abnormalities with the successive layers is given in Table 3 along with a summary of the full network is described in Table 4.

#### 4.1.1. Convolutional Blocks of CNN Model

The basic building component of the suggested architecture is the convolutional block, and each convolutional block consists of a convolutional 2D, a ReLU, and an average pooling 2D. The layer kernel initializer GlorotUniform-V2 is initialized to assign weights to the kernel of the layer. The gradient vanishing issue is fixed by the ReLU activation function, which also makes it possible for the network to learn and operate more quickly.

The convolutional 2D down-samples the picture and its spatial dimensions by averaging the values for each input channel throughout an input window (whose size is determined by pool size). The convolutional layers operate asymmetrically, building the features over time. Local patterns like edges, lines, and curves are recovered from the original layers, and local features are extracted based on these patterns as shown in Figure 5.

#### 4.1.2. Flattened Layer

A flattened layer is placed between the convolution layers and dense layers. Convolution layers work with tensor data types for input while dense layers require input in a one-dimensional format. A flattened layer vectorizes the feature map to feed it to dense layers, as depicted in Figure 6.

#### 4.1.3. Dropout Layer

Dropout layers turn nodes on and off to reduce the training time of the model and decrease the network complexity. Dropout randomly switches off nodes using probability distribution during each epoch, which prevents models from over-fitting. As a result, the model learns all the relevant features and completely prevents diverse features in each iteration.

### 4.2. Dense Block of Proposed Model

In this study, we use two dense blocks which contain multiple layers and the activation function implementation detail of each block is discussed in the next subsections.

#### 4.2.1. ReLU Activation

Activation functions are mathematical operations that decide whether output from perception is to be forwarded to the next layer. In short, they activate and deactivate nodes in a deep model. The activation function is used in the output layer to activate the node, which returns its label, which is then assigned to the image processed through the model. There are several activation functions. We used ReLU in hidden layers because of its simple and time-saving calculation. SoftMax, a probability-based activation function, is used for the output layer because our model is for multi-class classification.

#### 4.2.2. Dense Layer

The dense layer is also called a fully connected layer. This layer inputs a single vector and produces output based on its parameters. The images are identified and assigned a class label in these layers. The learning of the model takes place in fully connected layers via the back-propagation method. The number of trainable parameters of a model is determined based on the number of values used in each dense layer. SoftMax is used after a couple of layers, with the number of neurons equal to the number of classes [34]. The labels are one-hot encoding in multi-class classification, and only the positive class is present in the loss term.

## 5. Model Evaluations

The experiments were executed on a personal computer system equipped with two Intel Xeon 2687W v4 (3.0 GHz clock speed, 12 cores and 24 threads) CPUs, 64 GB RAM, 5 GB (NVIDIA) P2000 GPU (Graphical Processing Unit). The evaluation of the model was carried out by using the test set created from the splitting of the dataset before training the model. We employed different metrics to benchmark the performance of our model. The following evaluation metrics are extensively used to analyse the performance of the proposed deep CNN model for GI tract disorder detection:

### 5.1. Accuracy

Accuracy is the measure of total correct predictions out of total predictions obtained using the following expressions:(1)$$\begin{array}{c} Accuracy=\left(\frac{TP+TN}{TP+FN+FP+TN}\right) \end{array}$$
where, TP, TN, FN, and FP are True Positive, True Negative, False Negative, and False Positive values, respectively.

### 5.2. Precision

Precision is the ratio of correct positive predictions to total positive predictions and it is calculated using the following equation:(2)$$\begin{array}{c} Precision\;\;=\left(\frac{TP}{TP+FP}\right) \end{array}$$

### 5.3. Recall

Recall is also known as sensitivity score or true positive rate. It is the comparison of correct positive predictions to total actual correct positives. Recall is calculated using the following equation:(3)$$\begin{array}{c} Recall\;\;=\left(\frac{TP}{TP+FN}\right) \end{array}$$

### 5.4. F1-Score

Ideally, a value of 1.0 in precision and 1.0 in the recall is considered an ideal case for a classification model. F1-score is the harmonic mean of precision and recall. F1-score is unique in the sense that it plots its graph with a separate line for each class label. The F1-score is computed using the following equation:(4)$$\begin{array}{c} F1\;\;=\left(2\times \frac{Precision\;\times \;Recall}{Precision\;+\;Recall}\right) \end{array}$$

### 5.5. Receiver Operating Characteristics (ROC) Curve

An ROC curve is a graphical way to illustrate the possible connection between sensitivity and specificity for every possible cut-off for a combination of tests. The ROC curve graph is illustrated with the help of 1-specificity (on the x-axis) and sensitivity (on the y-axis). The 1-specificity is the false positive rate and sensitivity is the true positive rate, which can be obtained through the following expressions:(5)$$\begin{array}{c} TPR=\left(\frac{FP}{FP+FN}\right) \end{array}$$
(6)$$\begin{array}{c} FPR=\left(\frac{FP}{FP+TN}\right) \end{array}$$

### 5.6. Confusion Matrix

A confusion matrix is used to assess and calculate different metrics of a classification model. It provides the division of the number of all the predictions a model has made during the training or testing phase.

### 5.7. Loss Function

Loss functions calculate the mathematical difference between the predicted value and the actual value. For this research, we used a categorical cross-entropy algorithm for loss.
(7)$$\begin{array}{c} Loss\;\;=y\;-\;\bar{y} \end{array}$$
(8)$$\begin{array}{c} L_{CE}\;\;=\;-\;\sum_{n=1}^{k}\left(L_{i}\;log\;\left(p_{i}\right)\;\right) \end{array}$$
where ’L’ is the calculated loss of each class and ’P’ is the probability calculated by the Soft function.

## 6. Results and Discussion

In the next subsection, we analyse GIDD-Net with the recent deep network. The comparison of the proposed GIDD-Net with state-of-the-art deep networks is illustrated in Table 5. We selected the same parameters for all the networks to fairly examine their performance.

### 6.1. Accuracy Comparison against Other Models

We evaluated our proposed and recent models such as VGG16 and Efficient-Net B0 using the same dataset before and after balancing it through BL-SMOTE. Remarkable results are provided by the system with BL-SMOTE for the proposed model and other models. The proposed GIDD-Net with BL-SMOTE, GIDD-Net without BL-SMOTE, VGG16, and Efficient-Net B0 achieved accuracies of 98.9%, 92.9%, 91.84%, and 83.86%, respectively, using an imbalanced AD dataset as shown in Figure 7. This significant improvement in accuracy of GIDD-Net with BL-SMOTE can be seen in Figure 7.

### 6.2. AUC Comparison against Other Models

Our proposed model is a deep CNN-based GIDD-Net consisting of different blocks that is very effective at classifying the different GI interacting disorder classes, as discussed earlier in this paper. We compared GIDD-Net with state-of-the-art deep networks VGG16 and Efficient-Net B0 to validate our deep GIDD-Net. The first model reaches an AUC value of 98.35% using the same dataset, while the second Efficient-Net B0 evaluation AUC results are 98.8% using the same dataset. The proposed GIDD-Net with BL-SMOTE, and GIDD-Net without BL-SMOTE attained AUC values of 99.90% and 99.07% on the same datasets, as depicted in Figure 8. As a result of the above discussion, we noted that the performance of the proposed model remains better and more consistent in comparison with other models in the form of AUC.

### 6.3. Comparison with Other Models Using Precision

We evaluated our proposed and recent models like VGG16 and Efficient-Net B0 using the same dataset before and after balancing it through BL-SMOTE. Remarkable results are provided by the system with BL-SMOTE for the proposed model and other models. The proposed GIDD-Net with BL-SMOTE, GIDD-Net without BL-SMOTE, VGG16, and Efficient-Net B0 achieved precision values of 98.9%, 93.77%, 92.26%, and 86.59%, respectively. As shown in Figure 9, the proposed precision values achieved by the model are 93.77% and 98.9%, before and after balancing the number of samples using BL-SMOTE. As a result of the preceding discussion, we discovered that the presented model’s performance is better and more consistent than recent deep models in the form of precision.

### 6.4. Comparison of GIDD-Net with Recent Models Using Recall

Recall is determined by dividing the total number of positive samples by the number of positive samples accurately categorized as positive. The model’s ability to recognize positive samples is measured by recall. Higher recall values represent a greater number of positive samples found. The proposed GIDD-Net is compared using a recall curve with VGG16 and Efficient-Net B0, as depicted in Figure 10. The proposed model’s training accuracy reached 98.60%, while the validation obtained 96.70% accuracy, 99.82% AUC, and an F1-score of 98.1%. The proposed GIDD-Net with BL-SMOTE, GIDD-Net without BL-SMOTE, VGG16, and Efficient-Net B0 achieved recall values of 98.8%, 92.69%, 91.67%, and 82.29%, respectively, as shown in Figure 10. The result of the preceding discussion is that the presented models give a remarkable performance in the form of recall.

### 6.5. Comparison of Proposed CNN with Recent Models Using F1-Score

The input dataset is normalized in this suggested ADD-Net model, and the fundamental procedure of converting categorical data variables is delivered to the model utilizing the one-hot encoder. The BL-SMOTE technique is then used to correct the unbalanced dataset problem by oversampling the classes to balance the dataset. The proposed GIDD-Net with BL-SMOTE, GIDD-Net without BL-SMOTE, VGG16, and Efficient-Net B0 achieved F1-score values of 98.1%, 88.15%, 83.87%, and 73.41%, respectively, as shown in Figure 11. This significant improvement in F1-score of the proposed GIDD-Net with BL-SMOTE can be seen in Figure 11.

### 6.6. Loss Comparison with Recent Deep Models

Loss functions calculate the mathematical difference between the predicted value and actual value. For this research, we used a categorical cross-entropy algorithm for loss calculation. Optimization functions are backtracking algorithms that adjust the weights and biases of layers based on the value of the loss. However, the results are even more outstanding when the model is trained with up-sampled images. The proposed model’s training accuracy reached 98.60%, while the validation obtained 96.70% accuracy, 99.82% AUC, and an F1-score of 98.1%. The proposed GIDD-Net with BL-SMOTE, GIDD-Net without BL-SMOTE, VGG16, and Efficient-Net B0 achieved loss values of 0.0474, 0.2819, 0.4849, and 0.5130, respectively, as depicted in Figure 12. This significant improvement in the loss of the proposed GIDD-Net with BL-SMOTE is visible from Figure 12.

### 6.7. ROC Comparison with Recent Models

An ROC curve is used to analyze the performance of clinical tests and more specifically the accuracy of a classifier for binary or multi-classification. The Area Under the Curve (AUC) in an ROC curve is used to measure the usefulness of the classifier, where a greater AUC generally means greater usefulness of the classifier. We check the usefulness and accuracy of our proposed GIDD-Net using the ROC curve using the AD dataset with and without BL-SMOTE. The proposed GIDD-Net with BL-SMOTE and GIDD-Net without BL-SMOTE are compared using the ROC curve with VGG16 and Efficient-Net B0 on the same dataset. %. The proposed GIDD-Net with BL-SMOTE, GIDD-Net without BL-SMOTE, VGG16, and Efficient-Net B0 achieved ROC values of 0.9989, 0.9483, 0.9371, and 0.9160, respectively, as depicted in Figure 13. This significant improvement in the ROC of the proposed GIDD-Net with BL-SMOTE is visible from Figure 13.

### 6.8. Comparison with Other Models Using Extension of ROC for Multi-Class

ROC curves are commonly used in binary classification to investigate a classifier’s output. Binarizing the output is required to expand the ROC curve and ROC area to multi-class or multi-label classification. One ROC curve can be generated for each label; however, each element of the label indicator matrix can also be treated as a binary prediction (micro-averaging). The proposed GIDD-Net is compared using the extension of the ROC curve with VGG16 and Efficient-Net B0 as depicted in Figure 14. We can note that after balancing the dataset using the BL-SMOTE algorithm the AUC significantly improved for the proposed approaches in comparison with other models, as shown in Figure 14. This similar effect has also been noted in the AUC for all the classes of the proposed GIDD-Net with BL-SMOTE, and GIDD-Net without BL-SMOTE of class 0 (ampulla of vater), class 1 (angiectasia), class 2 (blood fresh), class 3 (blood hematin), class 4 (erosion), class 5 (erythema), class 6 (foreign body), class 7 (ileocecal valve), class 8 (lymphangiectasia), class 9 (Normal mucosa), class 10 (polyp), class 11 (pylorus), class 12 (Reduced mucosal), and class 13 (ulcer). These improvements in AUC prove the authenticity of the BL-SMOTE algorithm and feature selection of the GIDD-Net.

### 6.9. Comparison of GIDD-Net with Recent Models Using Confusion Matrix

In this proposed GIDD-Net, the input dataset is pre-processed using normalization and the essential process of converting the categorical data variables to be provided to the model using the one-hot encoder. Then, the BL-SMOTE algorithm is applied to resolve the imbalanced dataset issue that over-samples the classes to balance the dataset. We compared GIDD-Net with state-of-the-art deep networks VGG16 and Efficient-Net B0 to validate our deep GIDD-Net. Remarkable results are provided by the system with BL-SMOTE for the proposed model and other models, as depicted in Figure 15. To generate the CAM, the class gradient score is computed relative to the feature maps of the CNN layers. The preceding class gradient is related to the channel, ensuing the last CNN layer to generate a localization CAM displaying the important locations in the image that has a substantial effect on the deep model’s prediction as shown in Figure 16.

## 7. Conclusions

In medical imaging areas including image processing and analysis, CNN models can revolutionize deep image analysis by accelerating workflows and boosting productivity. These results demonstrate the applicability of the most recent pre-trained architectures VGG16 and Efficient-Net B0. In this study, we propose a robust CNN model with fewer parameters for early detection and classification of the most frequent GI disorders. Each convolutional block of adapted architecture is carefully constructed with several layers and is used to categorize GI tract infections in their early phase. To balance the number of samples for each class and address dataset imbalance issues, the BL-SMOTE technique is used to create new instances. The Grad-CAM technique shows a heat map of class activation to explain how the CNN layer functions. Our suggested deep model offers a remarkable 98.9% accuracy, 98.9% per cent precision, 98.8% sensitivity (recall), and an amazing AUC value of 99.8% percent. In the future, we will incorporate more trained architectures and improve transfer learning models to provide more desirable outcomes.

## Figures and Tables

**Figure 1 biomedicines-10-02195-f001:**
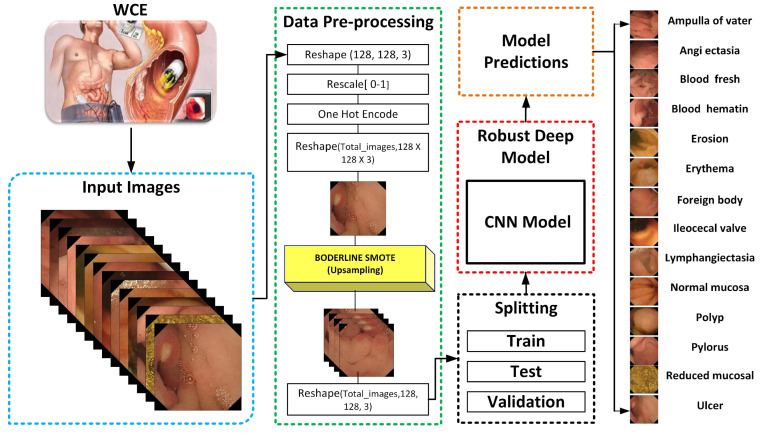
Work flow of the proposed classification model for WCE.

**Figure 2 biomedicines-10-02195-f002:**
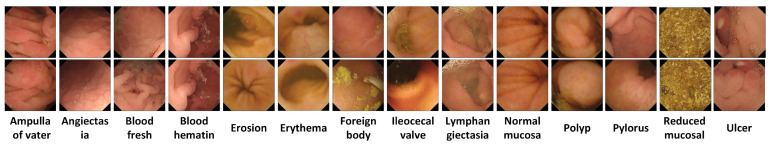
Original Kvasir-Capsule dataset samples extracted from WCE video frames with respect to each class label.

**Figure 3 biomedicines-10-02195-f003:**
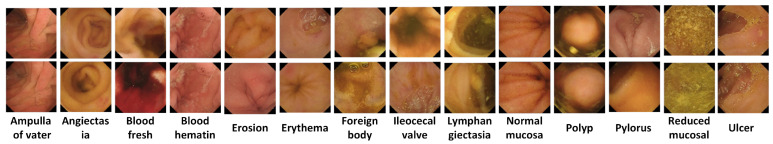
BL-SMOTE synthesizes image samples to handle the class imbalance problem.

**Figure 4 biomedicines-10-02195-f004:**
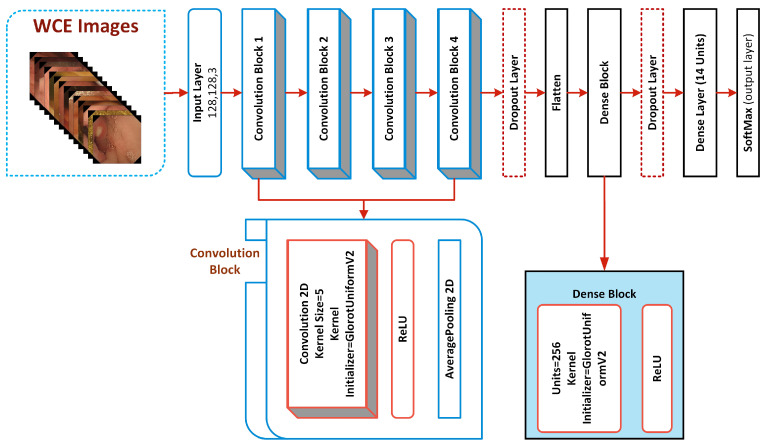
Detailed architecture of robust deep CNN to classify gastric disorders.

**Figure 5 biomedicines-10-02195-f005:**
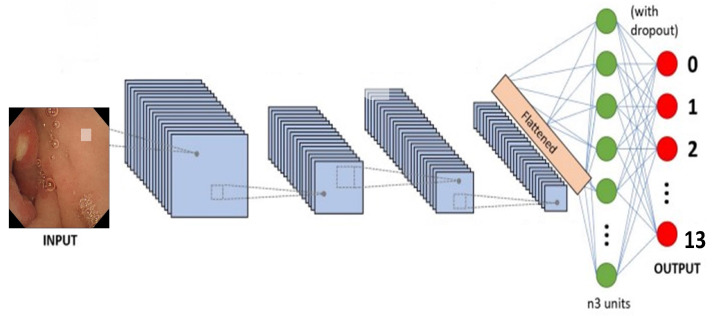
Generalized architecture of traditional CNN model.

**Figure 6 biomedicines-10-02195-f006:**
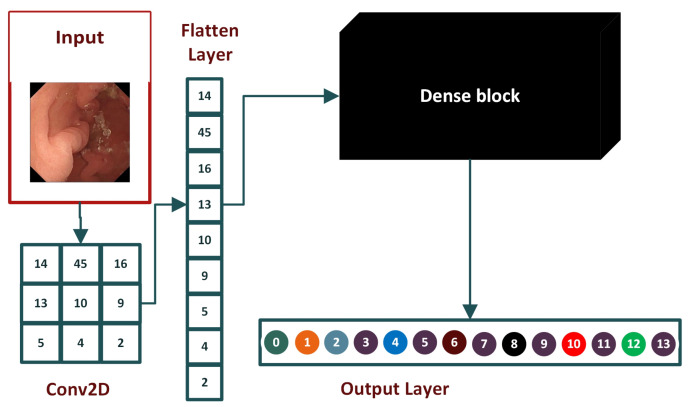
The base architecture of the flattened layer.

**Figure 7 biomedicines-10-02195-f007:**
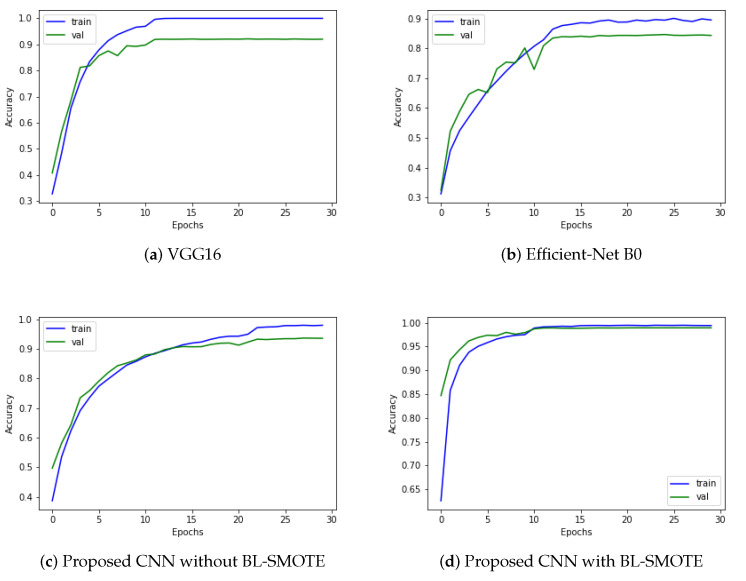
Significant improvement in accuracy scores with or without up-sampling of proposed model as compared to other state-of-the-art algorithms.

**Figure 8 biomedicines-10-02195-f008:**
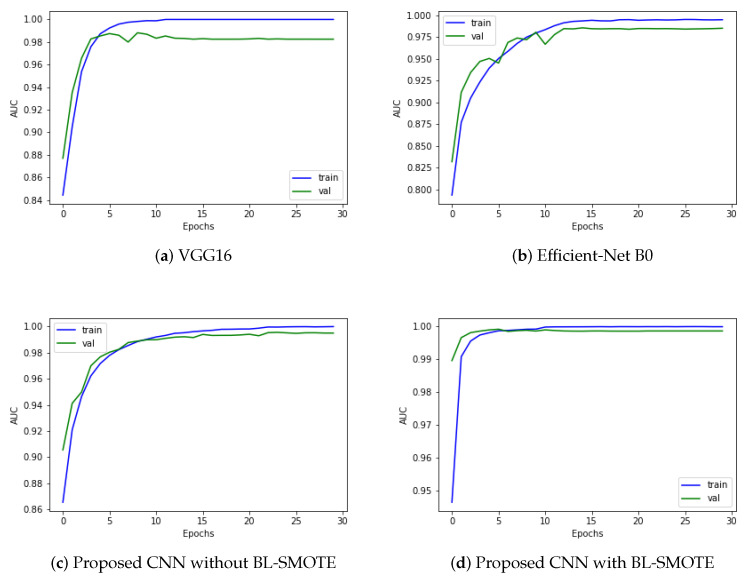
Area under the curve analysis to visualize multi-class classification results of (**a**) VGG16, (**b**) Efficient-Net B0, and (**c**,**d**) proposed deep CNN model with and without up-sampling.

**Figure 9 biomedicines-10-02195-f009:**
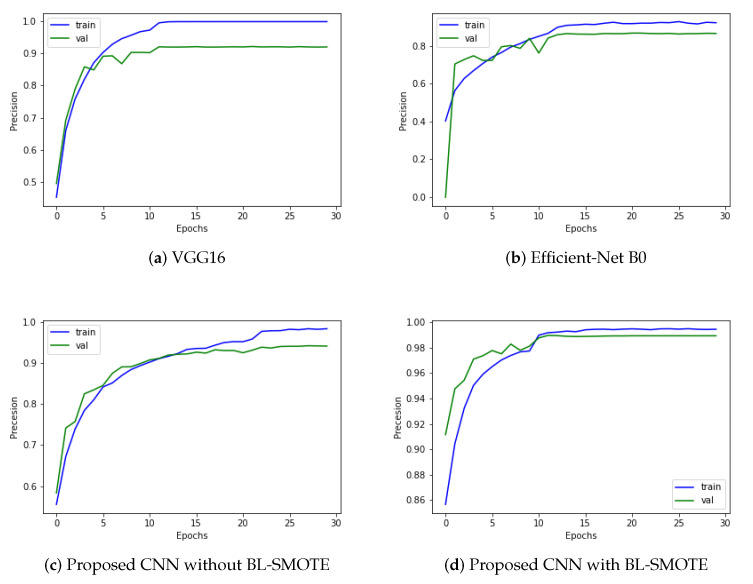
Precision results to provide an overview of the trade-offs between actual positive and positive predictive value for (**a**) VGG16, (**b**) Efficient-Net B0, and (**c**,**d**) DAD-Net with and without up-sampling.

**Figure 10 biomedicines-10-02195-f010:**
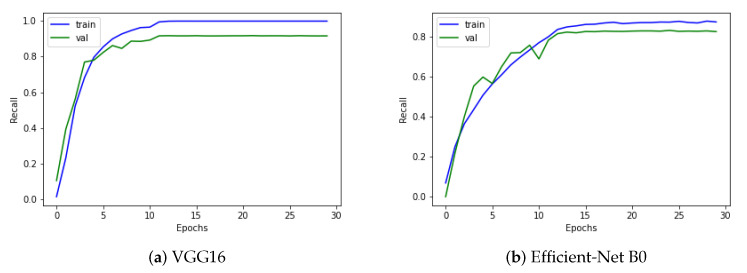
The recall analysis to provide an overview of the sensitivity between the actual positive and the positive predictive value for (**a**) VGG16, (**b**) Efficient-Net B0, and (**c**,**d**) DAD-Net with and without up-sampling.

**Figure 11 biomedicines-10-02195-f011:**
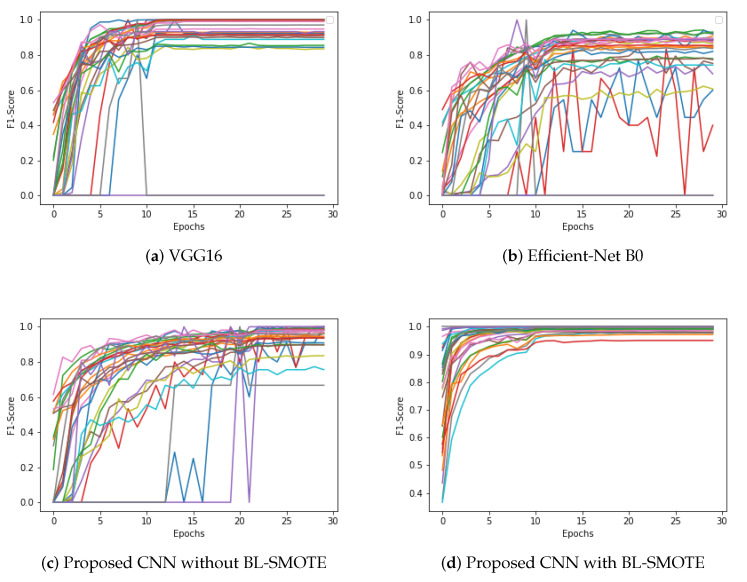
Calculate F1-score by taking the harmonic mean of a classifier’s precision and recall (**a**) VGG16, (**b**) Efficient-Net B0, and (**c**,**d**) proposed deep CNN model with and without up-sampling.

**Figure 12 biomedicines-10-02195-f012:**
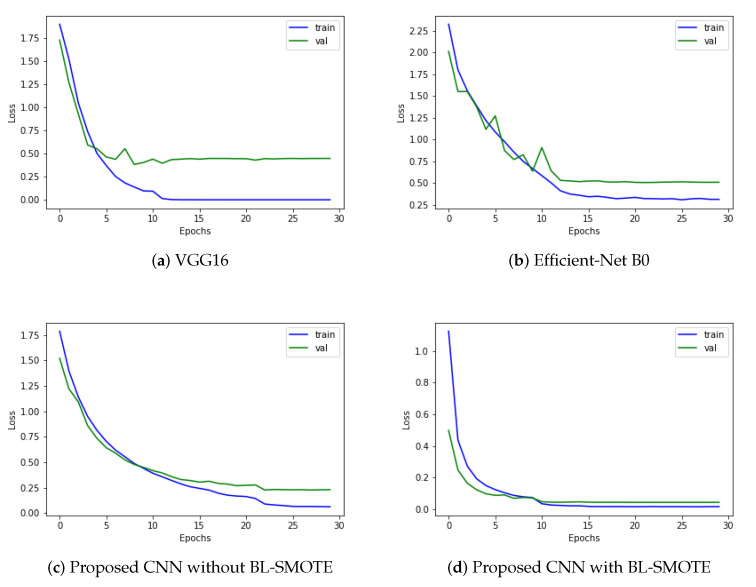
Training process loss of (**a**) VGG16, (**b**) Efficient-Net B0, and (**c**,**d**) proposed deep CNN model with and without up-sampling.

**Figure 13 biomedicines-10-02195-f013:**
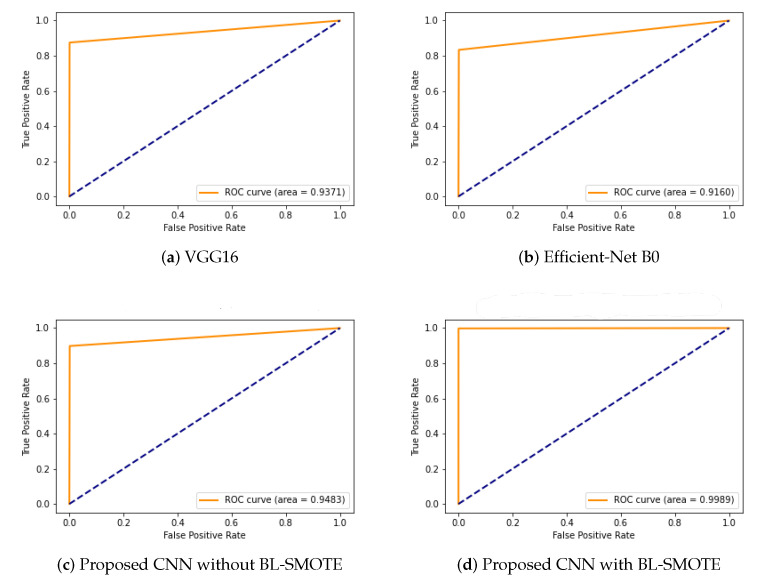
The ROC curve to compare the performance of (**a**) VGG16, (**b**) Efficient-Net B0, and (**c**,**d**) proposed deep CNN model with and without up-sampling.

**Figure 14 biomedicines-10-02195-f014:**
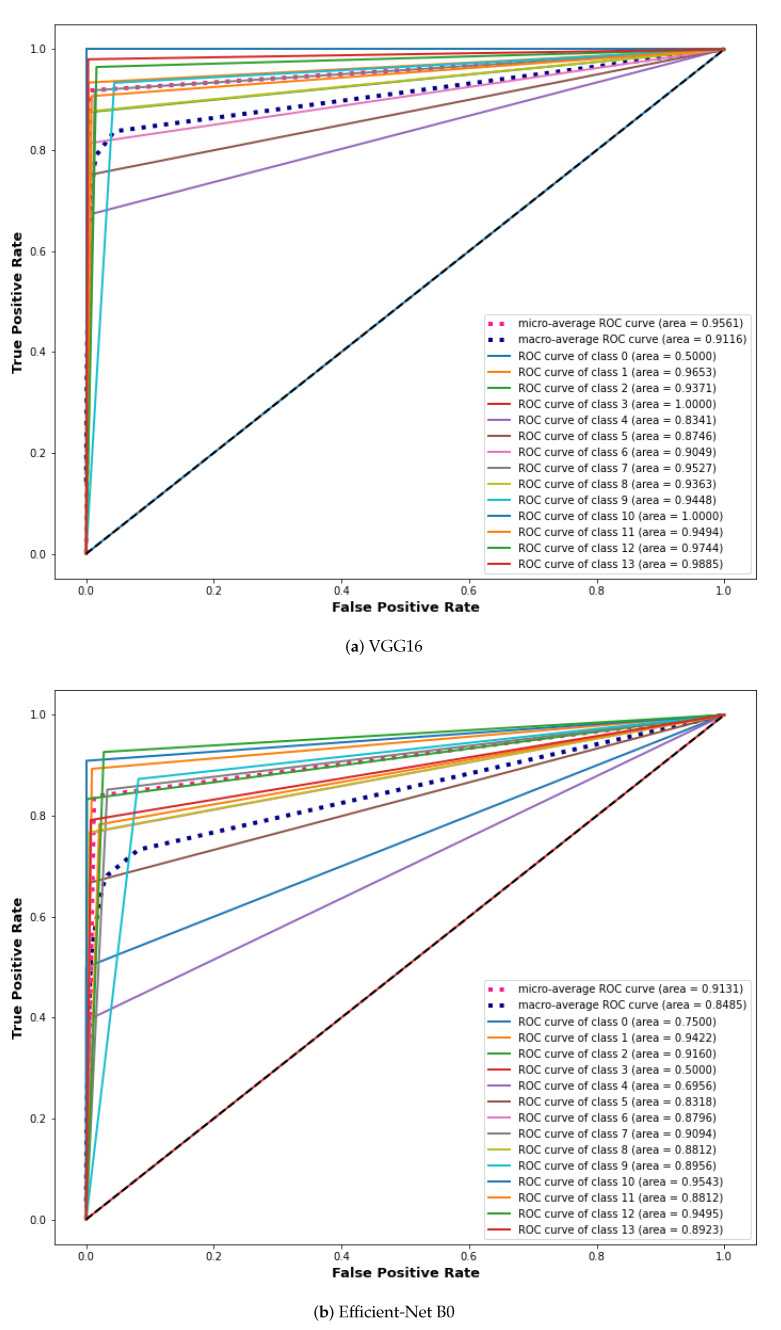
ROC curve analysis using extension receiver of (**a**) VGG16, (**b**) Efficient-Net B0, and (**c**,**d**) proposed deep CNN model with and without up-sampling.

**Figure 15 biomedicines-10-02195-f015:**
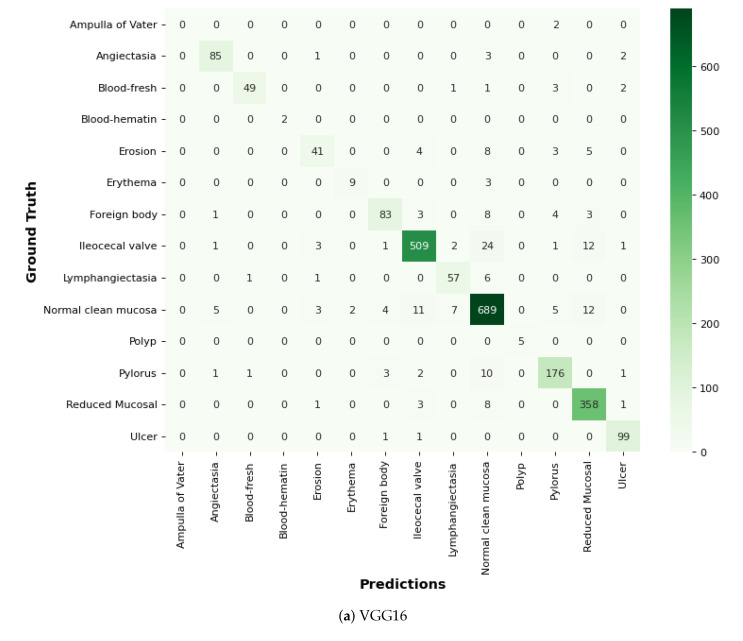
Comparison of the proposed model using confusion matrix.

**Figure 16 biomedicines-10-02195-f016:**
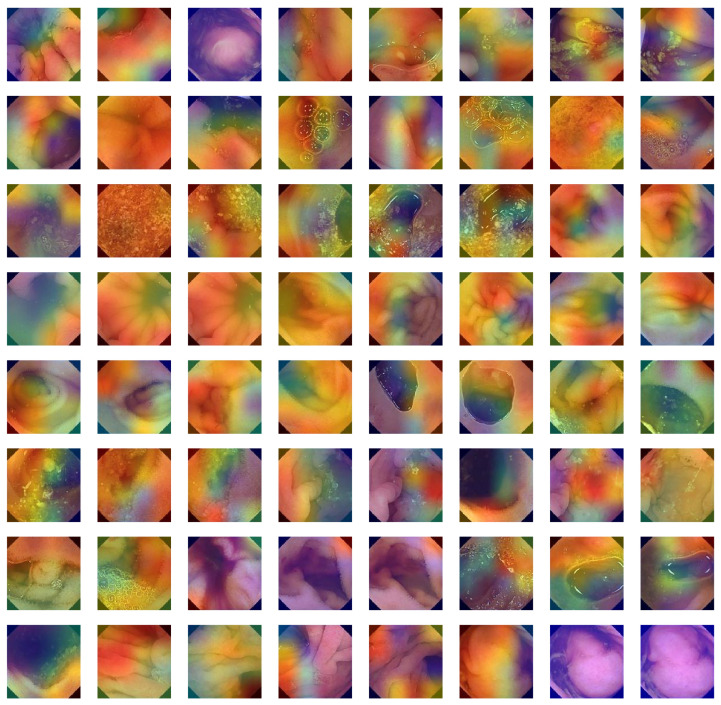
Grad-CAM visualization of GI tract abnormalities.

**Table 1 biomedicines-10-02195-t001:** A literature review of related studies on WCE image analysis.

Approach	Method	Accuracy	AUC	Precision	Recall	F1-Score
J. Yogapriya [10]	VGG16	96.3%	–	96.5%	96.3%	96.5%
	ResNet-18					
	Google-Net					
Prabhanantha Kumar [28]	ResNet-50	95.1%	–	95.6%	–	95%
See Wang [26]	HAnet	92.05%	97.2%	92.3%	91.6%	91.9%
	ResNet-34					
Abbas Biniaz [17]	CNN Model (SVD)	96%	–	91%	89%	90%
Jia-sheng [22]	CNN Model	97%	–	97.8%	–	96.5%
Subhashree Mohapatra [21]	CNN Model	93.6%	–	93.7%	93.6%	93.6%
Debesh Jha [29]	NenoNet-A	93.8%	–	93.3%	97.8%	93.6%
Husanbir Singh Pannu [30]	CNN Model	95%	–	91%	94%	92%

**Table 2 biomedicines-10-02195-t002:** Distribution of image samples in WCE dataset before up-sampling.

Sr. #	Class Name	No. of Images
0	Ampulla of vater	10
1	Angiectasia	866
2	Blood fresh	446
3	Blood hematin	12
4	Eroson	506
5	Erythema	159
6	Foreign body	776
7	lleocecal valve	4189
8	Lymphangiectasia	592
9	Normal clean mucosa	5927
10	Polyp	55
11	Pylorus	1529
12	Reduced mucosal view	2906
13	Ulcer	854

**Table 3 biomedicines-10-02195-t003:** Distribution of image samples in WCE dataset after up-sampling.

Sr. #	Class Name	No. of Images
0	Ampulla of vater	5927
1	Angiectasia	5927
2	Blood fresh	5927
3	Blood hematin	5927
4	Eroson	5927
5	Erythema	5927
6	Foreign body	5927
7	lleocecal valve	5927
8	Lymphangiectasia	5927
9	Normal clean mucosa	5927
10	Polyp	5927
11	Pylorus	5927
12	Reduced mucosal view	5927
13	Ulcer	5927

**Table 4 biomedicines-10-02195-t004:** List of total parameters used in proposed CNN model.

Model Summary		
**Layer Type**	**Output Shape**	**Parameters**
Input Layer	(None, 128, 128, 3)	0
Block01	(None, 62, 62, 32)	2432
Block02	(None, 29, 29, 64)	51,264
Block03	(None, 12, 12, 128)	204,928
Block04	(None, 4, 4, 256)	819,456
Dropout_1	(None, 4, 4, 256)	0
Flatten	(None, 4096)	0
Dense_1	(None, 256)	1,048,832
Dropout_2	(None, 256)	0
Dense_2	(None, 14)	3598
Output: SoftMax	(None, 14)	0
Total Parameters		2,130,510
Trainable Parameters		2,130,510
Non-Trainable Parameters		0

**Table 5 biomedicines-10-02195-t005:** Performance comparison of proposed CNN model and other state-of-the-art algorithms.

Architecture	Dataset	Accuracy	AUC	Precision	Recall	F1-Score
Proposed Model (With BL-SMOTE)	Kvasir-Capsule	98.7%	99.8%	98.7%	98.6%	98.7%
Proposed Model (Without BL-SMOTE)	Kvasir-Capsule	92.9%	99%	93.7%	92.7%	88.2%
Efficient-Net B0	Kvasir-Capsule	83.86%	98.4%	86.6%	82.3%	73.4%
VGG16	Kvasir-Capsule	91.84%	98.6%	92.3%	91.7%	83.87%

## Data Availability

The supporting data for the findings of this study are available from the corresponding author on reasonable request. While the source code will be publicly available at https://github.com/shahidzikria/WCE-Classification-by-handling-Imabalnce-problem-using-BLSMOTE.

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
