# Peer review of "A Robust Deep Model for Classification of Peptic Ulcer and Other Digestive Tract Disorders Using Endoscopic Images"

_biomedicines, 2022, doi:10.3390/biomedicines10092195_

Round 1

Reviewer 1 Report

The abstract needs modification. The need for this research has to be mentioned. As per  table 3 upscaling of images 10 images in the 0 case is increased to 5927 how this will affect the intermediate results of the classes?

Table 1 there is no need for the column imbalance handling. Introduction may have tree representation. what is reason behind to select VGG 16 instead of popular VGG 19? MCC, Jaccard Index and Kappa are to be analyzed. Please check the title for figure 11.d. What is need for ROC in this case? Explain. In table 5 The AUC is almost constant when compare with accuracy and other parameters what could be a reason for this phenomenon. The enhancement of references are needed. The conclusion needs correction in the content.

Author Response

Reviewer#1, Concern # 1: The abstract needs modification. The need for this research has to be mentioned. As per table 3, upscaling of images, ten images in the 0 case are increased to 5927 how this will affect the intermediate results of the classes?

Author response:  Yes, thank you for your valuable suggestion. The abstract was updated successfully by adding the need for the proposed research, and I attached the screenshots of the updated abstract. We revised the manuscript by modifying the abstract as follows:

Concern#01, part (2): As per table 3, upscaling of images, ten images in the 0 case are increased to 5927. How this will affect the intermediate results of the classes?

Author response: In the medical dataset, the big issue is the imbalance of the dataset. As we know, the accuracy of deep models is compromised due to the class imbalance problem of the dataset. Oversampling and under-sampling are two techniques to overcome this problem, so upscaling of class 0 allows equal participation of this class in model training. Practical model training archives a remarkable accuracy score. Table 5 clearly describes the performance comparison of the proposed model before and after upscaling. Accuracy in Fig. 7(c) and Fig. 7(d) shows the performance of the model with and without smote, respectively; graphs are also added below as Fig. 7(a) and Fig. 7(b). Moreover, F1-Score is the combination of precision and recall. With and without borderline SMOTE results of F1-Score are 98.1% and 88.15%, respectively. The big difference is due to each class’s comparison with other classes; when types are balanced, the issue is solved automatically, as clearly visible in Fig. 7(c) and Fig. 7 (d). For those classes having more number samples, the line of that class goes Up; for those not having many samples, like 0 class, the line is in the middle. Accuracy goes up after balancing the dataset. However, the upscaling of class 0 will affect the intermediate classes’ results by improving model training which will cause improved model accuracy.

Figure 7. Compared to other state-of-the-art algorithms, there is a significant improvement in accuracy scores with or without up-sampling of the proposed model.

Concern # 2: In table 1, there is no need for column imbalance handling. The introduction may have a tree representation. what is the reason behind selecting VGG 16 instead of the popular VGG 19? MCC, Jaccard Index, and Kappa are to be analyzed. Please check the title for figure 11.d. What is needed for ROC in this case? Explain. In table 5 The AUC is almost constant when compare with accuracy and other parameters which could be a reason for this phenomenon. The enhancement of references is needed. The conclusion needs correction in the content.

Author response: 

  1. Thanks for your kind suggestion. I update the manuscript by removing the imbalance handling column. This column shows that all existing models cannot handle the imbalance problem of the dataset. That’s why the accuracy of those models is compromised.
  2. This model shows a testing accuracy of 90.27%, whereas our model shows an accuracy of 92.99% of accuracy without smote. We compared it with most of the pre-trained models. However, we included the results that are suitable to add. You can check the performance of VGG19 in the graphs. VGG19 has parameters of 7,04,21,582 and VGG16 has 65,111,886 parameters. VGG19 and VGG16 comparison is added below, add evaluation reports of VGG16 and VGG19 below and mention their testing accuracies.
  3. MCC, Jaccard Index, and Kappa are not analysed because the selected evaluation metrics are frequently used in the literature to explore the model. Consequently, we choose the same evaluation metrics to study our model.

  1. We have updated the caption of 11(d)
  2. ROC curve is used to measure the performance summary of the classification model according to the classification threshold.
  3. AUC is a detailed class-wise performance measure of a classification model. That’s why the values of AUC remain high or may come with minor differences in both cases with or without up-sampling. Values of AUC typically, an excellent model has AUC near the one, which means it has a good measure of separability. A poor model with the worst has an AUC near 0, which implies a separability measure.

Reviewer 2 Report

This is an interesting study reporting the use of a synthetic oversampling technique BORDERLINE SMOTE (BL SMOTE) to improve accuracy in the analysis of endoscopic gastrointestinal images from the Kvasir-Capsule image dataset. Although the study is nicely devised, the article is plagued by some awkwardness of the English.

1. This manuscript needs to be edited by a native english speaker. Although the english is of high quality, there are minor problems with the grammar throughout.

2. The utility of this work is in the improved automated analysis of endoscopic images when compared to other deep learning architectures, however, interventional endoscopists are also getting interested in autonomous actions. What do the authors feel aboutthe feasibility of autonomous actions during endoscopy? How can their findings help bring about improvements in autonomous endoscopic actions?

Gumbs AA, Alexander F, Karcz K, Chouillard E, Croner R, Coles-Black J, de Simone B, Gagner M, Gayet B, Grasso V, Illanes A, Ishizawa T, Milone L, Özmen MM, Piccoli M, Spiedel S, Spolverato G, Sylla P, Vilaça J, Abu Hilal M, Swanström LL. Erratum: White paper: definitions of artificial intelligence and autonomous actions in clinical surgery. Art Int Surg 2022;2:120-1. http://dx.doi.org/10.20517/ais.2022.17

Wagner M, Bodenstedt S, Daum M, Schulze A, Younis R, Brandenburg J, Kolbinger FR, Distler M, Maier-Hein L, Weitz J, Müller-Stich BP, Speidel S. The importance of machine learning in autonomous actions for surgical decision making. Art Int Surg 2022;2:64-79. http://dx.doi.org/10.20517/ais.2022.02

3. Could the authors also make a quick comment regarding quantum computers? How do they foresee this technology impacting their future work?

Author Response

The authors response is attached below.

Round 2

Reviewer 1 Report

The  Jaccard  score analysis must be carried out and analyzed

Author Response

Please, find the attached response sheet.
